# Observation of elastic spin with chiral meta-sources

Weitao Yuan[1,2,3], Chenwen Yang[1,3], Danmei Zhang[1], Yang Long[1], Yongdong Pan[2], Zheng Zhong[2], Hong Chen[1], Jinfeng Zhao [2✉] & Jie Ren [1✉]

Directional routing of one-way classical wave has raised tremendous interests about spin-related phenomena. This sparks specifically the elastic wave study of pseudo-spin in meta-structures to perform robust manipulations. Unlike pseudo-spin in mathematics, the intrinsic spin angular momentum of elastic wave is predicted quite recently which exhibits selective excitation of unidirectional propagation even in conventional solids. However, due to the challenge of building up chiral elastic sources, the experimental observation of intrinsic spin of elastic wave is still missing. Here, we successfully measure the elastic spin in Rayleigh and Lamb modes by adopting elaborately designed chiral meta-sources that excite locally rotating displacement polarization. We observe the unidirectional routing of chiral elastic waves, characterize the different elastic spins along different directions, and demonstrate the spin-momentum locking in broad frequency ranges. We also find the selective one-way Lamb wave carries opposite elastic spin on two plate surfaces in additional to the source chirality.

[1] Center for Phononics and Thermal Energy Science, China-EU Joint Lab on Nanophononics, Shanghai Key Laboratory of Special Artificial Microstructure Materials and Technology, School of Physics Science and Engineering, Tongji University, Shanghai 200092, China. [2] School of Aerospace Engineering and Applied Mechanics, Tongji University, Shanghai 200092, China. [3]These authors contributed equally: Weitao Yuan, Chenwen Yang. ✉email: jinfeng.zhao@tongji.edu.cn; Xonics@tongji.edu.cn

Spin angular momentum (SAM) provides fundamental understanding of symmetrical and topological properties of physics[1–3], from quantum to classical aspects[4–6]. For electron, it allows people to realize robust one-way edge states propagating with spin-selected direction in what is known as topological insulator[7,8]. While in classical wave systems, an analogue degree of freedom called pseudo-spin is proposed by carefully designing the meta-structures, such as in electromagnetism[9–14] and acoustics[15–22], so that a time-reversal symmetry can be realized in the mathematical sense to implement the pseudo-spin-dependent topological phenomena. In particular, in the field of elastic topological metamaterials, the pseudo-spin has been synthesized by joining two degenerate degrees of freedom mathematically, such as using two inter-coupled vibrating oscillators[23], combing symmetric and antisymmetric elastic modes[24,25], exploiting double degenerated Dirac cones[26,27], and extracting elastic vortex feature with two valley degrees of freedom[28].

Different from pseudo-spin, the SAM of elastic wave has been theoretically uncovered from the elastic wave equation in conventional solids and describes the local rotation of displacement polarization[29]. The SAM gives rise to the quantum spin Hall effect (QSHE) for surface wave along the interface of conventional medium with the spin-momentum locking and unidirectional wave transport[1,29]. In contrast, the pseudo-spin relates more to the synthetization of different states like clockwise and anticlockwise energy flux in special structures or lattice systems[27,30]. As such, the edge modes related with pseudo-spin can be topological protected in their own states space. Recently, the spin-based on-chip magnetic devices and sensors[31,32] have been shown to contain spin transfer between homogeneous elastic and magnon systems without meta-structures, indicating that the elastic spin must play a significant role in the magnetic dynamics through elastic-magnetic SAM couplings. Yet, so far the direct experimental observation of elastic spin is still absent. Due to the large acoustic impedance, high operation frequency, and low vibration amplitude of elastic wave in solid materials, it is inevitably challenging to build an ultrasonic chiral source with sufficient radiation power, not even to mention the observation of rotating displacement polarization.

Here, by characterizing the displacement field and SAM on conventional metal plate upon the ultrasonic chiral meta-sources, we report the observation of intrinsic elastic spin and experimental demonstration of its basic properties, including the spin-resolved wave mode and the deterministic spin-momentum locked one-way propagation. The results advance the understanding of elastic spin and broadband spin-dependent properties in general solids, provide people more possibility and flexibility to design the on-chip elastic device[26–28], and pave a way for further revealing new phenomena of chiral elasticity[33–35], spin transfer[36], and conversion in coupled multi-physical systems[31,32].

## Results

**Principle of elastic spin and chiral meta-sources.** We first focus on chiral Rayleigh wave that exhibits tight elastic spin-momentum locking on the free surface of semi-infinite solid. The displacement vector of surface Rayleigh wave is elliptically polarized, whose SAM density is described by the elastic spin[29], as:

$$\mathbf{S} = \frac{\rho\omega}{2}\text{Im}[\mathbf{u}^* \times \mathbf{u}], \quad (1)$$

where $\rho$ is the density of elastic medium, $\omega$ denotes the circular frequency, $\mathbf{u} = \{u_x, u_y, u_z\}$ is the displacement vector, and $*$ represents the conjugate operation. It should be noted that the elastic SAM describes a local rotating polarization in time domain

rather than the vorticity of a displacement field along $x$ or $-x$ in space domain.

To show the spin-momentum locking from the point-view of elastic wave equation, we combine Eq. (1) and the equation of Rayleigh wave (see Method Eq. (8)), and we obtain the SAM of Rayleigh wave on the solid–air interface as:

$$\begin{aligned}\mathbf{S}_R &= \hat{\mathbf{z}}\frac{\rho\omega}{2}\text{Im}[u_x^* u_y - u_y^* u_x]\\ &= -\hat{\mathbf{z}}k_R\kappa_l\kappa_t\rho\omega A^2 \frac{(k_R^2 - \kappa_t^2)(k_R^2 + \kappa_t^2 - 2\kappa_l\kappa_t)}{(k_R^2 + \kappa_t^2)^2},\end{aligned} \quad (2)$$

where $u_x$ and $u_y$ are the displacements along $x$-axis and $y$-axis, respectively, $\hat{\mathbf{z}}$ is unit vector along $z$-axis, and $k_R$ is the wave number of Rayleigh wave. Here, $\kappa_l = \sqrt{k_R^2 - k_l^2}$ and $\kappa_t = \sqrt{k_R^2 - k_t^2}$ are the bulk wave numbers of longitudinal and transverse wave, respectively. Obviously, the sign of $\mathbf{S}_R$ is tightly correlated with the sign of $k_R$ at $y = 0$, demonstrating the spin-momentum locking of Rayleigh wave.

The existence of spin-dependent edge modes can also be confirmed by the specific spin Chern number of homogeneous solids. Considering a pair of basic elastic wave states, i.e., the normalized longitudinal wave $\mathbf{u}_l = (1, 0, 0)e^{ik_lx}$ and circular polarized transverse wave $\mathbf{u}_t = \frac{1}{\sqrt{2}}(0, i\sigma, 1)e^{ik_tx}$ where $k_l$ and $k_t$ are wave numbers of longitudinal and transverse waves, respectively, and $\sigma = \pm 1$. One can obtain the Berry curvatures of these two basic states as $F_l = \nabla_\mathbf{k} \times [-i(\mathbf{u}_l)^* \cdot (\nabla_\mathbf{k})\mathbf{u}_l] = 0$ and $F_t^\sigma = \nabla_\mathbf{k} \times [-i(\mathbf{u}_t^\sigma)^* \cdot (\nabla_\mathbf{k})\mathbf{u}_t^\sigma] = \sigma\frac{\mathbf{k}_t}{k_t^3}$, respectively. The topological Chern numbers are $C_l = \frac{1}{2\pi}\oint F_l d^2\mathbf{k} = 0$ and $C_t^\sigma = \frac{1}{2\pi}\oint F_t^\sigma d^2\mathbf{k} = 2\sigma$ for the longitudinal and transverse wave, respectively. The total Chern number is $C = C_l + \sum_{\sigma=\pm1} C_t^\sigma = 0$ due to the time-reversal symmetry, but the spin Chern number of elastic wave is $C_{spin} = C_{spin}^l + C_{spin}^t = 0 + \sum_{\sigma=\pm1}\sigma C_t^\sigma = 4$, leading to the existence of two pairs of edge modes with opposite spins[29]. Note that, these elastic surface modes are not robust against sharp corners or defects due to their trivial topological $Z_2$ invariant ($C_{spin}/2$ mod $2 = 0$). So that, these surface modes are lacking of backscattering immunity. Similar topological discussion is also presented in the quantum spin hall effect of light[1].

To excite chiral Rayleigh mode and observe this spin-momentum locking effect, we design an elastic spin source with four vibrating rods. As shown in Fig. 1a, a chiral elastic source whose elastic spin is positive (resp. negative) can only excite the left-going (resp. right-going) wave that has the positive (resp. negative) elastic SAM. This tight coupling between SAM and propagating direction of Rayleigh wave can be understood according to Rayleigh wave equation (see Methods, Eq. (8)), i.e. the positive (resp. negative) $\mathbf{S}$ along $z$-axis only supports negative (resp. positive) wave vector $\mathbf{k}$ along $x$-axis. Consequently, the one-way Rayleigh wave can be selectively excited by the elastic sources with chirality.

To construct such an effectively chiral elastic source in solids applicable for broad ultrasonic frequency regime, we elaborately design an array of sub-wavelength meta-sources with clockwise or anticlockwise phase shift, as shown in Fig. 1b, instead of adopting the macroscopic mechanical stirring in soft matters[34]. The chiral meta source constitutes four breathing rods labeled with Rod 1 to Rod 4. The displacement vectors around a single rod all point to the rod center or all away from the rod center, which allows the polarization of displacement vector to be always opposite between the inner and outer area. As such, the sign of SAM is opposite between the inner and outer area of the chiral meta-sources. Considering the elastic meta-sources couples with neighboring field through its outer area, we call the source whose elastic SAM

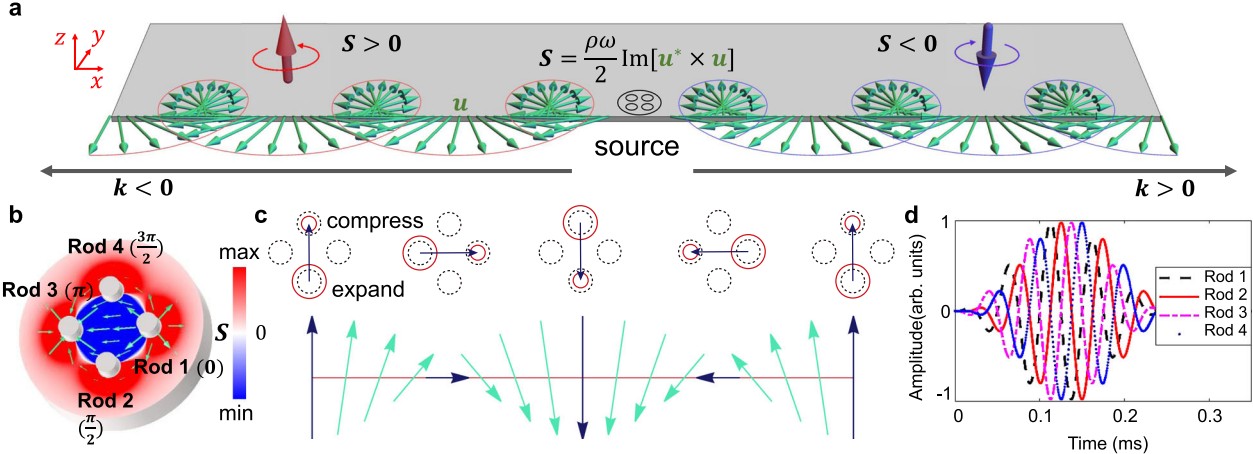

**Fig. 1 Schematic diagram of elastic spin and chiral meta-sources for surface elastic wave. a** Illustration of elastic spin of chiral Rayleigh wave. Here, the $x$–$z$ surface at $y = 0$ is seen as the interface between solid bulk and air, and the bulk size along $y$ axis is large enough to be seen as semi-infinite condition. The green vectors show the displacement polarization **u** for both the left-going (**k** < 0) and right-going (**k** > 0) Rayleigh wave on top of semi-infinite solid. The thick red/blue arrows denote the positive/negative SAM of surface wave, which describes the spiraling of displacement field. **b** The SAM distribution of the synthesized elastic spin-up source with input signals loaded on Rod 1 to 4, respectively. The red/blue color denotes the normalized positive/negative elastic SAM in the outer/inner area. The green arrows denotes the snapshot of displacement vector at a particular moment. **c** The time evolution (from left to right) of the displacement polarization at the center of the elastic spin-up source. During each period, the displacement vector between four rods rotates one cycle, as shown by the black arrows. **d** The input five-cycled tone burst signals for elastic spin-up source, with an incremental $\pi/2$-phase delay from Rod 1 to 4.

in outer area is positive as elastic spin-up source while the negative one as elastic spin-down source. To set rods into breathing vibration, we symmetrically fix two lead zirconate titanate piezoelectric ceramic (PZT) rings on two laterals of each rod (Supplementary Fig. 1a, b). Then five-cycled tone burst pulse signal, as shown in Fig. 1d, is simultaneously imposed on the PZT ring pair from Rod 1 to 4 in order. The input signal phase difference, being $\pi/2$ versus the central frequency of burst signals, is adjusted between Rod 1 to 4 in the clockwise or anticlockwise way to get elastic spin-down or elastic spin-up source, respectively. Notice that the choice of tone burst signal allows us to distinguish the initial wave package from the reflective ones, so as to observe the SAW locked transportation clearly.

**Observation of elastic spin for chiral Rayleigh wave**. Figure 2a shows the simulation results on the half-infinite plate with elastic spin sources. Obviously, the elastic spin-up/down source excites the Rayleigh wave along $-x/x$ direction, which agrees well with Eq. (2). Figure 2b shows the experimental setup whereby the chiral elastic meta-sources is implemented around center $(x, y) = (0, 1.2 \text{ cm})$. The aluminum plate has thickness 120 cm along $y$ axis, i.e., ~8 times of the largest Rayleigh wavelength in this work, to ensure the Rayleigh mode propagating on $x$–$z$ surface at $y = 0$. Figure 3a–d show the measured spatiotemporal pattern of $u_y$ and the mapped FFT component at each point around the central frequency 28 kHz. Obviously, the elastic spin-up source prefers to excite Rayleigh wave with negative **k**, while the elastic spin-down source mainly generates Rayleigh wave with positive **k**. These results are consistent with theory (see Methods), magnifying the tight coupling between the source spin and the propagating direction of Rayleigh wave, as well as the elastic SAM of Rayleigh wave itself.

Quantitatively, Fig. 3e presents the rectified amplitude of $|u_y|$ of the unidirectional wave upon the elastic spin-down and spin-up source on free $x$–$z$ surface ($y = 0$). Unambiguously, the larger $|u_y|$ occurs at $x < 0$ side when using elastic spin-up source (red) but shifts to $x > 0$ side with elastic spin-down source (blue). We then tune the central frequency of input signal, modify the phase difference, and

measure the out-of-plane displacement $u_y$ on free $x$–$z$ surface from $x = -55$ to $-25$ cm ($u_{y(x<0)}$) and $x = 25$ to 55 cm ($u_{y(x>0)}$) within a broad ultrasonic frequency range. Figure 3f shows that the rectification ratio of $|u_{y(x<0)}/u_{y(x>0)}|$ (red, elastic spin-up source) and its counterpart of $|u_{y(x>0)}/u_{y(x<0)}|$ (blue, elastic spin-down source) are always larger than 1 in the working frequency range.

Figure 4a, b then show the measured spectrum of $|u_y|$ with input signal frequency centered at 28 kHz, which are well matched to theoretical Rayleigh wave branch (red solid line). Figure 4a (resp. b) is obtained when excited with elastic spin-up (resp. down) source and presents higher energy density on the left (resp. right) branch, respectively. Similar phenomena are observed when input signals are tuned in the broad frequency range 21–37 kHz. It is worth emphasizing that here not only the source chirality but also the SAM of the chiral wave itself is tightly locked to the wave propagation direction.

We further measure the SAM of chiral Rayleigh wave to confirm the spin-momentum locking. To this end, we perforate small V-shape grooves in the surface of plate to obtain the out-of-plane displacement $u_y$ and the in-plane displacement $u_x$ (Supplementary Fig. 1c for details). Figure 4c, d show the measured temporal profiles of normalized $u_x$ and $u_y$ at position $x = -45$ cm (elastic spin-up source) and $x = 45$ cm (elastic spin-down source), respectively. In Fig. 4c, the phase of $u_x$ (blue solid line) at $x = -45$ cm is earlier than the one of $u_y$ (red dash line) before 1.6 ms. Due to wave reflection at the structure terminal, the signal after 1.6 ms gradually contains the information of reflected wave package. The wave vector **k** of reflected wave is opposite with the original one, so is the direction of SAM. Thus, the phase of $u_x$ is gradually later than the one of $u_y$ after 1.6 ms. The inset shows the anticlockwise rotation of displacement polarization **u** which corresponds to a positive **S**. The situation is reversed with elastic spin-down source excitation, where the phase of $u_x$ at $x = 45$ cm is later than that of $u_y$ while the rotation direction of polarization **u** is clockwise, resulting in negative **S**.

We would like to put it clear that the displacement polarization at $x = \pm 45$ cm is recorded by alternatively using two elastic spin sources for the convenience of illustration. Considering the measured unidirectional wave propagation and the obtained signs

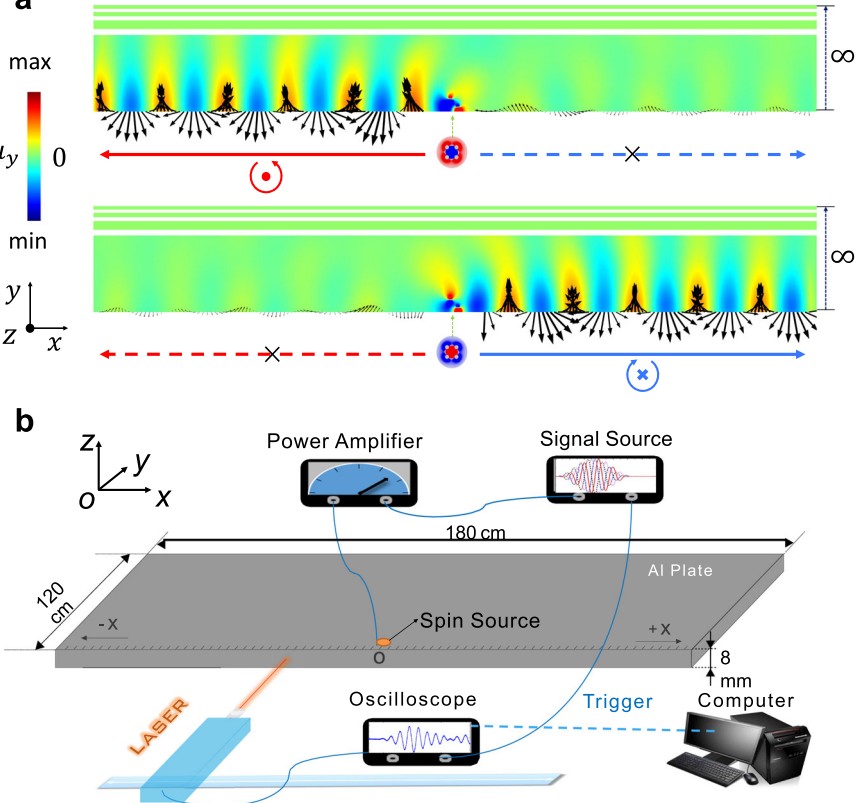

**Fig. 2 Simulation and experimental setup of elastic spin and chiral selection of Rayleigh mode. a** The simulation of spin-momentum locking in Rayleigh wave. By using elastic spin-up/down source, the Rayleigh wave will selectivity transport along -x/x-axis, respectively. The magnitude and direction of displacement polarization at the free boundary are shown with black arrows. **b** Schematic of measurement setup. The plate thickness (along y axis) is effectively infinite, much larger than the skin depth of Rayleigh wave in ultrasonic frequency range 21–37 kHz. Rayleigh wave is observed by characterizing the out-of-plane displacement $u_y$ on x–z surface.

of **S** at both $x < 0$ and $x > 0$ sides, we find that the opposite unidirectional wave carries opposite elastic spin and the dispersion of Rayleigh wave in Fig. 4a, b are spin-resolved in the broad frequency regime. The chiral selection of one-way propagation relies on the angular momentum matching between the source spin and the elastic spin of wave itself on particular direction, which clearly demonstrates the spin-momentum locking effect (see Methods).

**Observation of elastic spin for chiral Lamb wave.** We next turn to investigate the elastic spin and spin-momentum locking for the lowest-order antisymmetric Lamb ($A_0$) wave. Different from Rayleigh wave, $A_0$ wave contains a couple of opposite SAM on x–z surfaces $y = 0$ and $y = 6$ cm (see Methods, Eqs. (12) and (13)), e.g., the left-going wave possess negative SAM at the back x–z surface $y = 6$ cm due to the antisymmetric nature of $A_0$ wave, as shown in Fig. 5b. To demonstrate this point, we carried out measurement on both x–z surfaces where $u_y$ is recorded for both the left ($x = -55 - -25$ cm) and right ($x = 25 - 55$ cm) sides every 1 cm. When using then elastic spin-up source with central excitation frequency 14 kHz, Fig. 6a, b illustrate the measure spectrum of $|u_y|$ on the front and back x–z surfaces, respectively. Evidently, the components are very large for $A_0$ wave but very minor for the lowest-order symmetric Lamb ($S_0$) wave, which confirms the efficient generation of $A_0$ wave. The primary density-of-state spot overlaps on the left $A_0$ branch over its right counterpart, showing the SAM-dependent coupling between the elastic spin-up source and $A_0$ wave with negative **k** on surface $y = 0$. The mapping of normalized $u_y$ on x–z surfaces $y = 0$ and $y = 6$ cm in frequency domain are shown in Fig. 5a, c,

respectively, magnifying also the tight spin-momentum locking effect. These are well consistent with the numerical results in Fig. 5b and theoretical results derived from $A_0$ wave equation (see Methods, Eqs. (12) and (13)).

To observe the elastic spin in chiral $A_0$ wave, small V-shape grooves are also truncated on the x–z surfaces at $y = 0$ and $y = 6$ cm. As such, we can measure the SAM profiles from the temporal signals of displacements $u_x$ and $u_y$, shown in Fig. 6c, d. The displacement polarizations at $x = -45$ cm on the front and back x–z surfaces are shown by insets of Fig. 6c, d, respectively. The rotations of displacement polarization are the same in the excited wave on $y = 0$ and in the outer area of chiral source, being akin to the SAM matching between chiral source and Rayleigh wave mode. Furthermore, the measurement demonstrates the positive **S** at front x–z surface $y = 0$ and negative **S** at back x–z surface $y = 6$ cm, which confirms the opposite SAM between two x–z surfaces indicated by Eqs. (12) and (13). This phenomenon is absent in Rayleigh wave but unique for Lamb wave system.

Relevant results of chiral selective generation of right-going $A_0$ wave, by using the spin-down source, is demonstrated in Supplementary Fig. 2. Lastly but importantly, the chiral selective routing of $A_0$ wave is dependent on the elastic spin of wave itself, and occurs in a broad frequency range whenever using the elastic spin-up or spin-down source, as shown in Fig. 5d, e and Supplementary Fig. 3.

## Discussion

In conventional solid structures, we have experimentally demonstrated the intrinsic spin in elastic waves, by measuring the

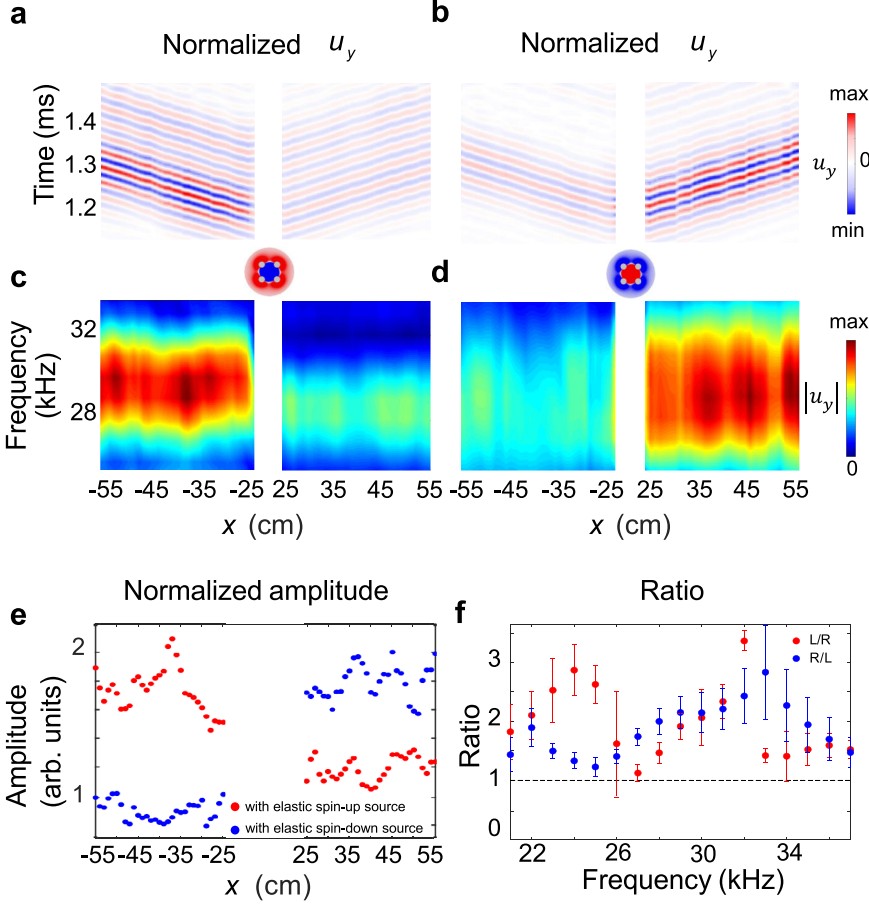

**Fig. 3 Experiment results about chiral transmission of Rayleigh wave.** The mapping of normalized $u_y$ measured by laser Doppler vibrometer along the segment ($x = -55$–$-25$ cm and $x = 25$–$55$ cm every 1 cm) in time (**a**, **b**) and frequency (**c**, **d**) domain, and they present the amplitude of Rayleigh wave with negative or positive **k**, respectively. The elastic SAM of excited source is positive (**a**, **c**) and negative (**b**, **d**), respectively. The central frequency of input signal is $f_c = 28$ kHz. **e** The rectified amplitude of the $|u_y|$ along x axis at 28 kHz, which all normalized to the $|u_y|$ recorded at point $x = -55$ cm with elastic spin-down source. The red and blue dots represent results with elastic spin-up and spin-down sources, respectively. **f** The average ratios of measured $|u_y|$ at $x = -55$–$-25$ cm and $x = 25$-55 cm, in a broad ultrasonic frequency regime. The red dots are for the $|u_{y(x<0)}/u_{y(x>0)}|$ when using elastic spin-up source while the blue dots stands for $|u_{y(x>0)}/u_{y(x<0)}|$ with the elastic spin-down source. The bars at each frequency are defined as the average ratio plus and/or subtract standard deviation of ratios derived at every group of points. The difference between two rectification ratio profiles comes from the imperfection of chiral sources during the manual installation.

SAM according to the rotating displacement polarization in Eq. (1). The chiral selective excitation of unidirectional wave propagation has been observed in experiments for both Rayleigh and Lamb wave systems without the help of any meta-structure, but by using the elaborately designed elastic chiral meta-sources.

In particular, the tight spin-momentum locking is confirmed by the fact that the locally rotating displacement polarization of the surface wave field itself determines the propagating direction. Moreover, we have observed the opposite elastic spin carried by the one-way Lamb wave on two-side surfaces in additional to the source chirality. These results agree well with the theoretical predictions and numerical simulations, in broad ultrasonic frequency ranges. Note that the present chiral elastic source, by elaborating specific phase on each PZT and rod from delaying signals (Supplementary notes D and E, and ref. [29]), is also generalized to study the acoustic spin[37], selective routing of near-field acoustics[38] and optics[39]. Besides, one may alternatively construct different chiral meta-source, by naturally adjust the underlying meta-structure[40] or multi-beam superposition to cope with different scenarios.

The experimental observation of elastic spin demonstrates the validity and feasibility of intrinsic spin angular momentum in

elasticity, which offers a promising platform for future investigation on integrative spin physics among electron[41], phonon[42], photon[4], and magnon[31,32,36], and provides new perspectives and means to the integrative on-chip surface-acoustic-wave devices that have shown the great potential in quantum acoustics[43,44].

## Methods

In continuum isotropic elastic media, the displacement filed can be represented by potential fields as:

$$\mathbf{u} = \nabla\psi + \nabla \times \boldsymbol{\Psi}, \tag{3}$$

where $\psi$ is the scalar potential function of longitudinal displacement field, and $\boldsymbol{\Psi}$ is the vector potential function of transverse displacement filed. Under 2D simplification, the z-component of displacement filed is $u_z = 0$, and the elastic wave equation can be written as follows:

$$\nabla^2\psi = \frac{1}{c_l}\frac{\partial^2\psi}{\partial t^2},$$

$$\nabla^2\Psi_z = \frac{1}{c_t}\frac{\partial^2\Psi_z}{\partial t^2}, \tag{4}$$

$$\frac{\partial\psi}{\partial z} = \Psi_x = \Psi_y = 0,$$

where $c_l$ and $c_t$ stand for the elastic wave speed of longitudinal and transverse waves, respectively. We set the free surface as x–z plane (y = 0) for which the stress

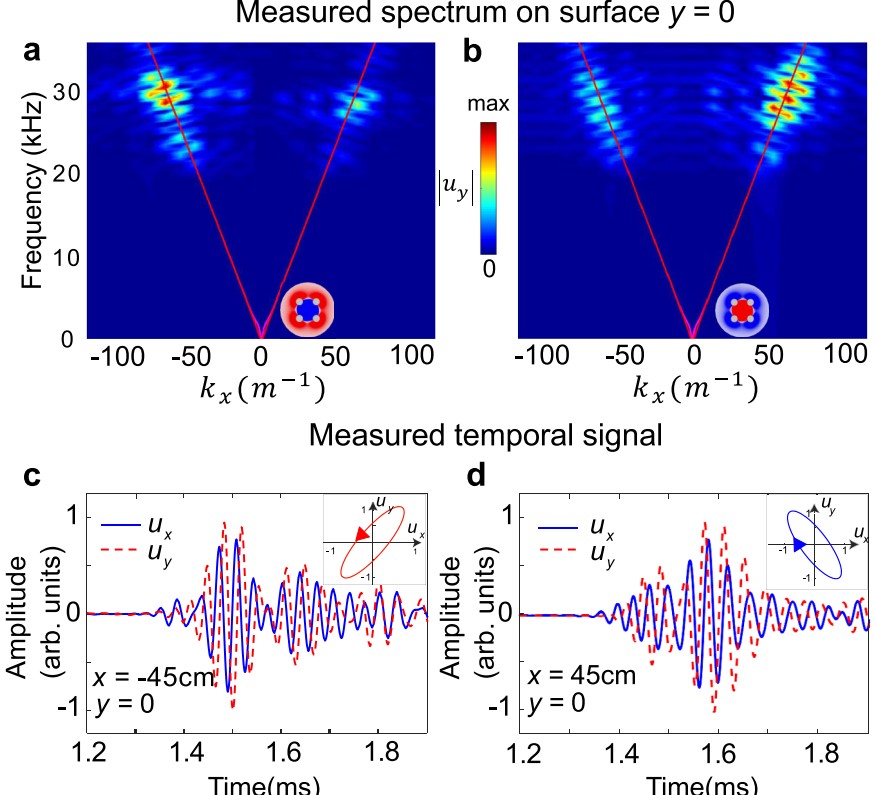

**Fig. 4 The spin-momentum locking of Rayleigh wave. a, b** Measured spectrum of Rayleigh wave on the $x$–$z$ surface $y = 0$ cm with the elastic spin-up (**a**) or elastic spin-down source (**b**) with $f_c = 28$ kHz, correspondingly. The red solid line is the theoretical dispersion for Rayleigh wave branch on semi-infinite aluminum solid. The red and blue circular arrows show the elastic spin direction of the left- and right-going waves. **c, d** The time evolution of $u_x$ (blue solid line) and $u_y$ (red dash line) measured at $x = -45$ cm (**c**) and $x = 45$ cm (**d**) when using elastic spin-up or spin-down source, respectively, normalized to the maximum amplitude of $u_y$ at each effective point. By extracting FFT components, i.e., amplitude and phase information, from displacement signals $u_x(t)$ and $u_y(t)$ at particular frequency, the elastic SAM is obtained in experiment. The insets show the anti-clockwise (**c**) and clockwise (**d**) rotation of displacement polarization $\mathbf{u} = (u_x, u_y)$.

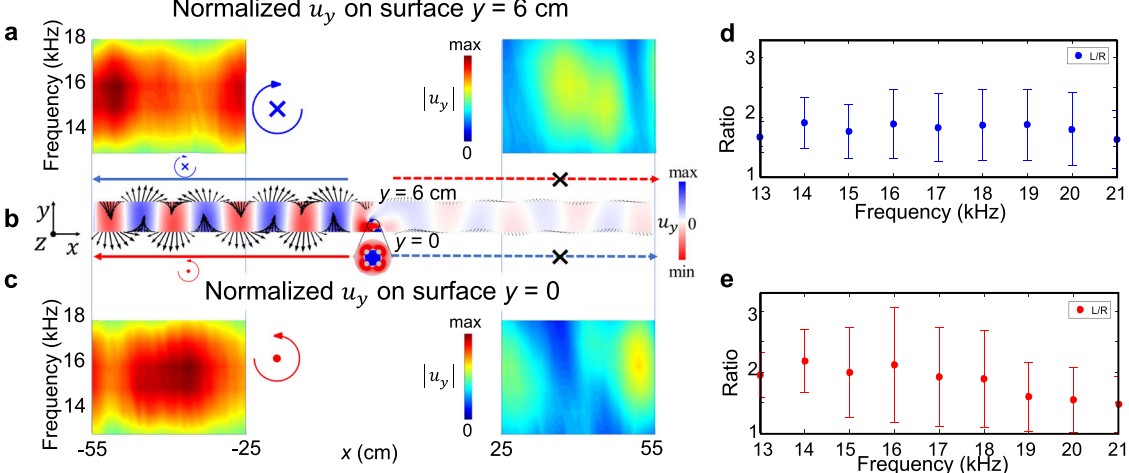

**Fig. 5 Experimental observation of chiral transmission of antisymmetric Lamb wave. a, c** The mapping of normalized $u_y$ measured along $x = -55$–$-25$ cm and $x = 25$-55 cm every 1 cm on back $x$–$z$ surface $y = 6$ cm and front $x$–$z$ surface $y = 0$ in frequency domain, respectively. Leftward unidirectional propagation is clearly observed in a broad frequency range. Here a 6 cm thick aluminum plate that occupies $y = 0$-6 cm along $y$ axis (depth direction) is chosen as the experimental sample, as shown in Supplementary Fig. 1a. And the chiral source is implemented at $(x, y) = (0, 1.2$ cm), near the front $x$–$z$ surface $y = 0$. **d, e** The average ratios of measured $|u_y|$ at $x = -55$–$-25$ cm and $x = 25$-55 cm, in a broad ultrasonic frequency regime. The dots are for the $|u_{y(x<0)}/u_{y(x>0)}|$ with elastic spin-up source on $x$–$z$ surface $y = 6$ cm (**d**) and $y = 0$ (**e**). **b** Simulated unidirectional $A_0$ wave upon the chiral source excitation propagates mainly towards left side with opposite elastic SAM on two parallel $x$–$z$ surfaces at $y = 0$ and $y = 6$ cm, which confirms the experimental observations.

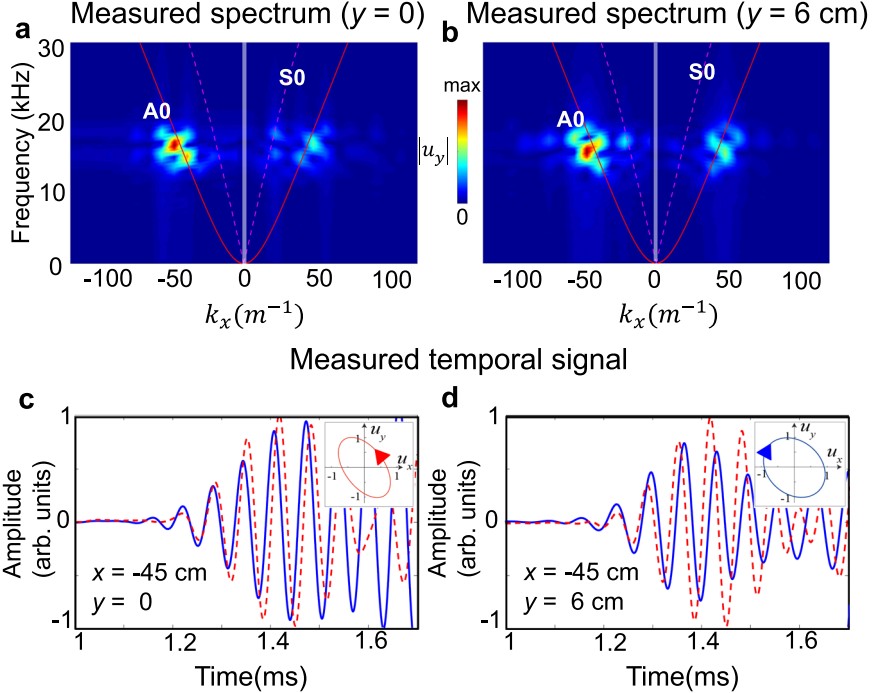

**Fig. 6 Experimental observation of the spin-momentum locking of antisymmetric Lamb wave. a, b** Measured spectrum of the chiral Lamb waves on front $x$–$z$ surface $y = 0$ and back $x$–$z$ surface $y = 6$ cm, respectively, upon the elastic spin-up source. Red solid and magenta dash lines stand for the theoretical dispersion of $A_0$ and $S_0$ wave branches in a 6 cm thick aluminum plate, respectively. Both densities of state significantly locate on the left $A_0$ wave branch, indicating the chiral selecitve excitation of one-way $A_0$ mode. **c, d** The time evolution of $u_x$ (blue solid line) and $u_y$ (red dash line) measured at $x = -45$ cm on $x$–$z$ surfaces $y = 0$ and $y = 6$ cm, respectively, when using elastic spin-up source. The insets show their corresponding clockwise and anti-clockwise rotation of displacement polarization **u** at frequency 14 kHz, for left-going wave. As such, the positive SAM of chiral source couples with the positive elastic SAM at front $x$–$z$ surface, which leads to the left-going $A_0$ wave.

$\tau_{yy}$, $\tau_{xy}$ and $\tau_{yx}$ are:

$$\tau_{yy} = \lambda\left(\frac{\partial^2\psi}{\partial x^2} + \frac{\partial^2\psi}{\partial y^2}\right) + 2\mu\left(\frac{\partial^2\psi}{\partial y^2} - \frac{\partial^2\Psi_z}{\partial x\partial y}\right) = 0,$$
$$\tau_{xy} = \tau_{yx} = \mu\left(2\frac{\partial^2\psi}{\partial x\partial y} - \frac{\partial\Psi_z}{\partial x} + \frac{\partial\Psi_z}{\partial y}\right) = 0,$$

(5)

Here, $\lambda$ and $\mu$ are Lame constants.

According to Eq. (4), the potential function of Rayleigh wave can be written as[45]:

$$\psi_R = Ae^{-\kappa_l y}e^{i(k_R x - \omega t)},$$
$$\Psi_{Rz} = Be^{-\kappa_t y}e^{i(k_R x - \omega t)}.$$

(6)

$\omega$ is the circular frequency, $A$ and $B$ are wave amplitude of longitudinal and transverse waves, respectively. $k_R$ is the $x$-component of Rayleigh wave vector. $k_l$ and $k_t$ are the wave numbers of longitudinal and transverse waves in bulk solid, respectively, whereby the $\kappa_l = \sqrt{k_R^2 - k_l^2}$ and $\kappa_t = \sqrt{k_R^2 - k_t^2}$. Typically, Rayleigh wave decays exponentially along $y$-axis.

Considering the free boundary condition at $y = 0$, together with Eqs. (5) and (6), the ratio $B/A$ can be written as:

$$\frac{B}{A} = \frac{2i\kappa_l k_R}{\kappa_t^2 + k_R^2}.$$

(7)

By accounting for Eqs. (3), (6) and (7), the displacement field of Rayleigh wave in $x$–$y$ plane can be expressed as:

$$u_x = ik_R\psi_R - \kappa_t\Psi_{Rz}$$
$$= iAk_R\left(e^{-\kappa_l y} - \frac{2\kappa_l \kappa_t}{k_R^2 + \kappa_t^2}e^{-\kappa_t y}\right)e^{i(k_R x - \omega t)},$$
$$u_y = -\kappa_t\psi_R - ik_R\Psi_{Rz}$$
$$= -A\kappa_l\left(e^{-\kappa_l y} - \frac{2k_R^2}{k_R^2 + \kappa_t^2}e^{-\kappa_t y}\right)e^{i(k_R x - \omega t)}.$$

(8)

Through the combination of Eqs. (1)[29] and (8), one can obtain the SAM of Rayleigh wave at $y = 0$, which is shown in Eq. (2).

Now consider a plate that occupies from $y = -d/2$ to $y = d/2$. The potential functions of antisymmetric Lamb wave can be written as[45]:

$$\psi_{A_0} = A\sinh(\kappa_l y)e^{i(k_{A_0} x - \omega t)},$$
$$\Psi_{A_0 z} = B\cosh(\kappa_t y)e^{i(k_{A_0} x - \omega t)},$$

(9)

where the hyperbolic function $\sinh(\kappa_l y)$ (resp. $\cosh(\kappa_t y)$) ensure that $u_x$ (resp. $u_y$) is opposite (resp. same) between surface $y = -d/2$ and $y = d/2$. $A$ and $B$ are wave amplitude of longitudinal and transverse wave, respectively. $k_{A_0}$ is the $x$-component of wave vector of $A_0$ mode. $k_l$ and $k_t$ are the wave numbers of longitudinal and transverse waves in bulk medium, and they are $\kappa_l = \sqrt{k_{A_0}^2 - k_l^2}$ and $\kappa_t = \sqrt{k_{A_0}^2 - k_t^2}$.

Then the displacement $u_x$ and $u_y$ can be expressed as:

$$u_x = ik_{A_0}\psi_{A_0} + \kappa_t\tanh(\kappa_t y)\Psi_{A_0 z},$$
$$u_y = \kappa_l\coth(\kappa_l y)\psi_{A_0} - ik_{A_0}\Psi_{A_0 z},$$

(10)

Being similar to Rayleigh wave, the stress $\tau_{yy}$, $\tau_{xy}$ and $\tau_{yx}$ vanish on the free boundary at $y = -d/2$ and $y = d/2$. We then obtain the ratio $B/A$ of antisymmetric Lamb wave by solving Eqs. (5), (9) and (10):

$$\frac{B}{A} = \frac{2i\kappa_l k_R\cosh\left(\frac{\kappa_l d}{2}\right)}{(\kappa_t^2 + k_R^2)\cosh\left(\frac{\kappa_t d}{2}\right)}.$$

(11)

By accounting for Eqs. (1), (10) and (11), the SAM of $A_0$ wave propagating in this plate is:

$$\mathbf{S}_{A_0} = -\hat{z}k_{A_0}\kappa_l\rho\omega A^2\frac{k_{A_0}^2 - \kappa_t^2}{k_t^4}\cosh\left(\frac{\kappa_l d}{2}\right)\left[k_t^2\sinh\left(\frac{\kappa_l d}{2}\right)\right.$$
$$\left. -2\kappa_l\kappa_t\cosh\left(\frac{\kappa_l d}{2}\right)\tanh\left(\frac{\kappa_t d}{2}\right)\right]$$

(12)

at $y = -d/2$, but becomes:

$$\mathbf{S}_{A_0} = \hat{z}k_{A_0}\kappa_l\rho\omega A^2\frac{k_{A_0}^2 - \kappa_t^2}{k_t^4}\cosh\left(\frac{\kappa_l d}{2}\right)\left[k_t^2\sinh\left(\frac{\kappa_l d}{2}\right)\right.$$
$$\left. -2\kappa_l\kappa_t\cosh\left(\frac{\kappa_l d}{2}\right)\tanh\left(\frac{\kappa_t d}{2}\right)\right]$$

(13)

at $y = d/2$. Clearly, $\mathbf{S}_{A_0}(y = -d/2) = -\mathbf{S}_{A_0}(y = d/2)$. The sign of $\mathbf{S}_{A_0}$ is tightly

correlated with the sign of $k_{A_0}$, which shows the spin-momentum locking of $A_0$ wave. Notice that the choice of $y$ from $-d/2$ to $d/2$, instead of the case from 0 to $d$ in both experiment and simulation, is just for the convenience of presenting the spin-momentum locking of $A_0$ wave as shown in Eqs. (12) and (13), while the physical relationship keeps unchanged.

Regarding the simulation, we use 2D elastic model in simulation software to illustrate the elastic spin-momentum locking for both Rayleigh and Lamb wave systems. The numerical meta-source is the same as the four-point source shown in Fig. 1b. For example, the simulation results when using the elastic spin-up and/or down excitation in Rayleigh wave are shown in Fig. 2a. Although the profile of displacement polarization on both sides are similar, the propagation direction is different. Hence, the rotation direction of local displacement polarization are opposite between the left and right sides. As such, the elastic spin-up (resp. elastic spin-down) source only excite Rayleigh mode in left (resp. right) side, while the wave mode in the other side is forbidden due to the mismatching in SAM.

## Data availability
The data that support the findings of this study are available from the corresponding author upon reasonable request.

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

## Acknowledgements
This work is supported by the National Natural Science Foundation of China (Nos. 11935010, 11775159, and 12172256), the Shanghai Science and Technology Committee (Nos. 18JC1410900 and 20ZR1462700), and the Opening Project of Shanghai Key Laboratory of Special Artificial Microstructure Materials and Technology.

## Author contributions
These two authors W.Y. and C.Y. contributed equally to this work. W.Y., Y.L., J.Z., and J.R. designed the chiral meta-source and experimental setups. W.Y., C.Y., D.Z., Y.P., and J.Z. performed the experimental measurements. C.Y., Y.L., and J.R. derived the theory. W.Y., C.Y., and D.Z. carried out the numerical simulations. Z.Z., H.C., J.Z., and J.R. conceived the project. All the authors contributed to discussion, interpreting the data, and the writing.

## Competing interests
The authors declare no competing interests.
