## [Peer Review File · Nature Communications]

REVIEWER COMMENTS

Reviewer #1 (Remarks to the Author):

The authors describe the observation of chiral elastic wave propagation, analogous to topological spin modes, in Aluminum plates supporting Rayleigh and Lamb waves. In both systems, they observe the unidirectional propagation, governed by different elastic spins along different directions.

The experimental setup is clever and simple: it relies on uniform plates driven by a group of four pillars with ring piezo-elements. The paper is well written and the experimental observations are convincing. I support the publication of the paper in Nature Communications, provided the authors could address the following points:

1. I understand the analogy between the chiral Rayleigh propagation and SAM. However, can the author truly demonstrate the topological nature of their system?
2. Most of the literature cited in the context of other systems demonstrating elastic topological metamaterials are discrete (i.e., rely of the design and arrangement of a finite number of cells). However, the solution proposed here is in a continuum. It would be interesting to see how the theoretical description of the observed propagation differs from earlier demonstrations.
3. The authors describe the Rayleigh propagation mathematically in the Methods section. However, it would be interesting to connect this derivation with the mathematics of topological properties in physics.
4. Also, for completeness, the authors should consider also including, in the methods section, a similar theoretical discussion for Lamb waves.
5. How and why was the burst signal selected as an excitation for the system? What happens if one changes the pulse duration, cycle numbers and envelope of the driving signal?
6. What causes the significant "leakage" of energy in the direction opposite to the expected propagation (see Fig. 2b, second and third panel, for example)?
7. Why did the authors change the normalization signal for selected figures (e.g., w.r.t. position 45 and/or 55)?

Reviewer #2 (Remarks to the Author):

The manuscript "Observation of elastic spin with chiral meta-sources" by Yuan et al. experimentally demonstrates the presence of elastic spin waves, Rayleigh and Lamb waves, generated by preparing a chiral elastic source. This work directly follows from the original theoretical work by Long et al. (PNAS 115, 9951), which points out the role played by intrinsic spin of elastic waves (locally rotating displacement polarization) in the propagation of waves.

The manuscript impressively provides a direct observation of Rayleigh modes on the surface of an aluminum (Al) plate, its spin momentum locking relation, dispersion, and the relative phase of displacement confirming its elliptical polarization. Moreover, the authors made the observation of chiral Lamb waves which exist on both sides of the Al plate, which further confirms the use of the chiral elastic source to generate chiral elastic spin waves that have potential applications to a broader area of condensed matter science including magnetic and photonic devices.

The manuscript is well written with the sets of figures convincingly supporting the claimed observations. I believe it has enough value and novelty to be published in Nature Communications.

To improve the readability for researchers from different background, I think the following points can be updated either in the main script or Supplemental Material:

1. Line 58-59. In the sentence above Eq.(1), it is correctly named 'the elastic spin'. Wherever

possible, the term 'elastic spin' should be used to distinguish it from 'spin' in other contexts.

2. Line 91,95. What does the PZT stand for?

3. Line 127-130. The ratio of the amplitudes between the left- and the right- going wave is about 2, which is not as large as one would expect. What factors in the experiment limit the ratio?

4. Fig.2(e). The elliptic polarization of displacement and its direction are well seen for a time interval 2-3 msec. For later time intervals, the polarization seems to be reversed. Please discuss what is being observed at a later time.

5. Line 182-185. Please explain why the chiral elastic source on one side of the Al plate generates A_0 , but not S_0 Lamb wave.

6. Please include the derivation of Eq.(2) from the superposition of longitudinal and transverse waves to compose Rayleigh waves.

7. Please include the derivation of Eq.(3) and related references.

Reviewer #3 (Remarks to the Author):

In the submission " Observation of elastic spin with chiral meta-sources", by Yuan et al., the authors make a clear observation of intrinsic elastic spin in Rayleigh-Lamb wave systems, without involving the artificial metamaterial structures and synthesized pseudospin. The intrinsic elastic spin is an important concept emerging quite recently and is attracting much attention in the community. Yet, direct experimental measurement of this new concept in elastic systems is still lack. The authors delicately realized a chiral elastic source, which works in a broad ultrasonic working frequency, and made a clever setup to measure the locally rotating displacement polarizations to overcome the technical challenging. To the best of my knowledge, the present manuscript reports the very first experimental measurement of the intrinsic spin of elastic waves and the first experimental realization of the chiral elastic source in broadband ultrasonic regime. I think the work will eventually add a nice piece to the textbook of elasticity/phononics. Overall, the writing is clear and I enjoy reading it. The results and analysis are convincing, both in experiment and in theory, and for sure will advance the field of "Chiral elasticity/phononics". Therefore, I am more than happy to recommend its publication in Nature Comm. But to further improve the manuscript before its final acceptance, I think there are some issues need to be addressed, as listed in below:

Main comments/suggestion:

1、 The difference between intrinsic spin and pseudo-spin should be one of the focuses of main text, but this is only briefly mentioned in abstract and introduction. Not enough. I think it's better to provide comprehensive discussions about the difference between intrinsic spin and pseudo-spin, as well as the advantage of intrinsic elastic spin in application prospect in more details. Most importantly, this is also a good chance (for the convenience of readers) to clarify the importance of the physical elastic spin and how it demarcates from the mathematical pseudo-spin, since the pseudospin and pseudospin-momentum locking have been widely studied in metamaterials, for example the valley pseudo-spin and vortices in Nature Physics 13, 369 (2017), PRB 95, 174106 (2017) and reference therein.

2、 The spin of chiral elastic meta-source is excited by four rods, as shown in Fig.1(b). As this source is not exactly a point source, the properties of source will be associated with the geometry size and wave length, and I assume the source will become invalid when the excitation frequency is high

enough. Is that true? If it is, how to determine the upper frequency limit of the source? I believe this is important since, in the application of such a chiral source, the working frequency range must be considered. Moreover, just out of curiosity, is it possible to enlarge the ratio of one-way propagation (possible to be perfect? And how?) shown in Fig.2(c) by optimizing the chiral meta-source, either experimentally or theoretically? Relevant discussions in AIP Advances 6, 085007 (2016) about vortices source may be helpful.

3、 The content of the current manuscript is very condensed. I'd like to suggest the authors present more details, for example:

a) The experiment setup details of Rayleigh wave are not clearly stated, for instance, the geometry size of the solid plate in Fig.2(a). Especially, the length of the plate along y-axis. According to the demonstration of chiral meta-source in Fig.1 and the eq.2 in methods, I think the theoretical model is based on half-infinite condition, which means that the length of the aluminum plate used in the experiment is required to ensure that the experiment can fit the simulation correctly. Also, the geometry and setup of chiral elastic source used for Rayleigh experiments are not well stated. More experimental details need be included in the revision.

b) The theoretical results about the elastic spin of Rayleigh and Lamb wave in Method are also very important results, which I strongly suggest move to the main text. It should be helpful to understand the intrinsic elastic spin and how it is tightly locked to the directionality. Moreover, the numerical simulation of Rayleigh wave spin and its spin-momentum locking should be helpful for the readers to understand the experimental measurements. Therefore, I suggest the authors move this part from the supplementary material to the main text.

Minors:

- 1、 When mention the surface in main text, please mark the surface with axis (surface x-z, e.g.) to avoid misunderstanding.
- 2、 It's better to provide the vibrating phase of each rod in the left panel of Fig.1(b).
- 3、 I think "measured spectrum" is a more proper title of Fig.2(d), 3(a) and 3(h).
- 4、 I recommend that the style of Fig. 3(b) and (g) remain consistent with the style of Fig. 2(e).

REVIEWER COMMENTS

The authors would like to sincerely thank all three reviewers for their valuable comments and suggestions. We have revised the manuscript (in red color) based on the reviewers' suggestions. The point-to-point response is as follows.

Reviewer #1:

1. I understand the analogy between the chiral Rayleigh propagation and SAM. However, can the author truly demonstrate the topological nature of their system?

Author: Thanks for your valuable concern about topological nature of our system. Our system is based on the spin angular momentum (SAM) of elastic wave where the topological properties are associated with the momentum-dependent polarization profile, so called spin-momentum locking, [Science 348, 1448 (2015); PNAS 115, 9951 (2018)]. Most importantly, the topological nature of our system can be extracted from the general elastic wave equations. Due to the nonzero spin Chern number for the elastic wave in homogenous isotropic solid ($C_{spin} = 4$) [PNAS 115, 9951 (2018)] and the bulk-edge correspondence, there are two pairs of propagation modes on the solid-air interface with opposite SAM. Meanwhile, because of the topological Z_2 invariant of elastic wave is trivial ($\nu = C_{spin}/2 \bmod 2 = 0$), these surface modes are lacking of backscattering immunity.

To clarify the topological nature of our system, we have added the description in the revised main text as

“The existence of spin-dependent edge modes can also be confirmed by the specific spin Chern number of homogeneous solids. Considering a pair of basic elastic wave states, i.e. the normalized longitudinal wave $\mathbf{u}_l = (1, 0, 0)e^{ik_l x}$ and circular polarized transverse wave $\mathbf{u}_t = (0, i\sigma, 1)e^{ik_t x}$ where \mathbf{k}_l and \mathbf{k}_t are wave numbers of longitudinal and transverse waves respectively and $\sigma = \pm 1$. One can obtain the Berry curvatures of these two basic states as

$$F_l = \nabla_{\mathbf{k}} \times [-i(\mathbf{u}_l)^* \cdot (\nabla_{\mathbf{k}})\mathbf{u}_l] = 0 \text{ and } F_t^\sigma = \nabla_{\mathbf{k}} \times [-i(\mathbf{u}_t^\sigma)^* \cdot (\nabla_{\mathbf{k}})\mathbf{u}_t^\sigma] = \sigma \frac{\mathbf{k}_t}{k^3} \text{ respectively.}$$

The topological Chern numbers are $C_l = \frac{1}{2\pi} \oint F_l d^2 \mathbf{k} = 0$ and $C_t^\sigma = \frac{1}{2\pi} \oint F_t^\sigma d^2 \mathbf{k} = 2\sigma$ for the longitudinal and transverse wave, respectively. The total Chern number is $C = C_l + \sum_{\sigma=\pm 1} C_t^\sigma = 0$ due to the time-reversal symmetry, but the spin Chern number of elastic wave is $C_{spin} = C_{spin}^l + C_{spin}^t = 0 + \sum_{\sigma=\pm 1} \sigma C_t^\sigma = 4$, leading to the existence of two pairs of edge modes with opposite spins [29]. Note that, these elastic surface modes are not robust against sharp corners or defects due to their trivial topological Z_2 invariant ($C_{spin}/2 \bmod 2 = 0$). So that, these surface modes are lacking of backscattering immunity. Similar topological discussion is also presented in the quantum spin hall effect of light [1].”

2. Most of the literature cited in the context of other systems demonstrating elastic topological metamaterials are discrete (i.e., rely of the design and arrangement of a finite number of cells).

However, the solution proposed here is in a continuum. It would be interesting to see how the theoretical description of the observed propagation differs from earlier demonstrations.

Author: Thanks for noticing this point. Our continuum system relies on the SAM of elastic wave which has been uncovered from the general elastic wave function [PNAS 115, 9951 (2018)]. The SAM of elastic wave that we discussed is an intrinsic physical quantity which describes the angular momentum contribution from the chiral polarization of the elastic wave. In contrast, the pseudo-spin is a synthetic and mathematical concept in topological models, which is highly dependent on the concrete structures or lattice systems (i.e, Honeycomb lattice) and usually only holds around some high symmetry points in the Brillouin zone (i.e, valley points). There is NO universal pseudo-spins for all topological models and our discussed system, the continuous isotropic solid. There is no concepts for pseudo-spins in our discussed continuous isotropic solid systems.

Furthermore, the elastic SAM results in nontrivial spin Chern number as $C_{spin} = 4$, while the trivial topological Z_2 invariant, so that the primary features are the spin-momentum locking and unidirectional wave transport, but no robust against backscattering, similar with the QSHE of light [Science 348, 1448 (2015)]. However, the discrete elastic topological metamaterials can provide nontrivial topological Z_2 invariant, result in robust edge modes against backscattering.

According to your suggestion and the discussion above, we add the following sentences at beginning of 2nd paragraph of main text:

“Different from pseudo-spin, the SAM of elastic wave has been theoretically uncovered from the elastic wave equation in conventional solids and describes the local rotation of displacement polarization [29]. The SAM gives rise to the quantum spin Hall effect (QSHE) for surface wave along the interface of conventional medium with the spin-momentum locking and unidirectional wave transport [1,29]. In contrast, the pseudo-spin relates more to the synthetization of different states like clockwise-anticlockwise energy flux in special structures or lattice systems [27,30]. As such, the edge modes related with pseudo-spin can be topological protected in their own states space. Recently, the spin-based on-chip magnetic devices and sensors [31,32] have been shown to contain spin transfer between homogeneous elastic and magnon systems without meta-structures, indicating that the elastic spin must play a significant role in the magnetic dynamics through elastic-magnetic SAM couplings.”

3. The authors describe the Rayleigh propagation mathematically in the Methods section. However, it would be interesting to connect this derivation with the mathematics of topological properties in physics.

Author: Thanks for your constructive suggestion. As we mentioned in the previous question, due to the bulk-edge correspondence, the existence of SAM dependent surface wave modes on the edge are confirmed by the nonzero spin Chern number. We have derived the nontrivial spin

Chern number in the former questions which implies the existence of two pairs of edge modes with opposite spins.

To show these surface modes that contains the spin-momentum locking as predicted by the spin Chern number, we have given a clear derivation about the SAM of surface wave on the edge, and moved the equation of SAM of Rayleigh wave from the method to the revised main text as:

“To show the spin-momentum locking from the point-view of elastic wave equation, we combine Eq. (1) and the equation of Rayleigh wave (See Method Eq. (8)), and we obtain the SAM of Rayleigh wave on the solid-air interface as:

$$\begin{aligned} \mathbf{S}_R &= \hat{\mathbf{z}} \frac{\rho\omega}{2} \text{Im}[u_x^* u_y - u_y^* u_x] \\ &= -\hat{\mathbf{z}} k_R \kappa_l \rho \omega A^2 \frac{(k_R^2 - \kappa_t^2)(k_R^2 + \kappa_t^2 - 2\kappa_l \kappa_t)}{(k_R^2 + \kappa_t^2)^2} \end{aligned} \quad (2)$$

where u_x and u_y are the displacements along x-axis and y-axis, respectively. $\hat{\mathbf{z}}$ is unit vector along z-axis, k_R is the wave number of Rayleigh wave. Here, $\kappa_l = \sqrt{k_R^2 - \kappa_t^2}$ and $\kappa_t = \sqrt{k_R^2 - \kappa_l^2}$ are the bulk wave numbers of longitudinal wave and transverse wave, respectively. Obviously, the sign of \mathbf{S}_R is tightly correlated with the sign of k_R at $y=0$, demonstrating the spin-momentum locking of Rayleigh wave.”

4. Also, for completeness, the authors should consider also including, in the methods section, a similar theoretical discussion for Lamb waves.

Author: According to this suggestion, we have given more detailed derivations of the SAM of both Rayleigh and Lamb waves in Section Methods.

5. How and why was the burst signal selected as an excitation for the system? What happens if one changes the pulse duration, cycle numbers and envelope of the driving signal?

Author: The choice of burst signal comes from practical aspects in experiment: first, the boundary reflection is almost unavoidable; secondly, the sample size cannot be physically infinite. The adoption of widely used 5-cycled burst signal features several advantages, including the separation of initial wave package from the reflective wave package, and the prohibition of unexpected vibration. We also give discussion on the influences caused by the pulse duration, cycle number, and envelope of the driving signal.

According to your question, we add the details of burst signals to Supplementary as a new section *“The selection of 5-cycled tone burst pulse signal for wave generation”*.

Also, in main text, we revise the relevant sentence in paragraph before Section *“Observation of elastic spin for chiral Rayleigh wave.”* into *“Then five-cycled tone burst pulse signal, as shown in Fig. 1(d), is simultaneously imposed on the PZT ring pair from Rod 1 to 4 in order.”*. Besides, we add a new sentence at the end of this paragraph *“Notice that the choice*

of tone burst signal allows us to distinguish the initial wave package from the reflective ones, so as to observe the SAW locked transportation clearly.”

6. What causes the significant “leakage” of energy in the direction opposite to the expected propagation (see Fig. 2b, second and third panel, for example)?

Author: This significant “leakage” of energy in opposite direction must be ascribed to the meta-source. In this work, we used a meta-source in experiment to mimic the point-like chiral source in simulation due to the experimental limitations. In constructing this meta-source, there are many factors that can affect the efficiency of unidirectional wave propagation or the significant “leakage” of energy in opposite direction. For example, through simulation, we find that the geometric parameters d , r_1 , L , l , h , and load on PZT rings, install errors can all affect the unidirectional propagation.

In viewing of your question, we have added revisions to Supplementary Information as a new section “*Factors affecting the sub-wavelength chiral source*” with Supplementary Fig. 5.

7. Why did the authors change the normalization signal for selected figures (e.g., w.r.t. position 45 and/or 55)?

Author: The change of the normalization signal for selected figures is just for the convenience of describing physical phenomena. In the new Fig 3(e) (or left panel of previous Fig. 2(c)), we show the variation of u_y amplitude when $x < 0$ and $x > 0$, and the amplitude is normalized to the smallest value at the leftmost point ($x = -55$ cm, with spin-down source). In the new Figs. 4(c) and 2(d) (previous Fig. 2(e)), we show the phase comparison between u_x and u_y , while the measured u_x and u_y were previously normalized to their maximum value separately.

In view of your question and to avoid confusion, the u_x and u_y are now normalized to the maximum value of u_y at each effective point in the revised manuscript. In the caption of Fig. 4, the “*normalized to the maximum amplitude of themselves*” is replaced by “*normalized to the maximum amplitude of u_y at each effective point*”. Similar revisions are done in Figs. 6(c) and (d).

Reviewer #2:

1. Line 58-59. In the sentence above Eq.(1), it is correctly named 'the elastic spin'. Wherever possible, the term 'elastic spin' should be used to distinguish it from 'spin' in other contexts.

Author: According to this suggestion, we have made revisions on the manuscript by replacing related “spin” with “elastic spin” where is possible.

2. Line 91,95. What does the PZT stand for?

Author: PZT stands for lead zirconate titanate piezoelectric ceramics. We revise the first “PZT” in main text to be “*lead zirconate titanate piezoelectric ceramics (PZT)*”.

3. Line 127-130. The ratio of the amplitudes between the left- and the right- going wave is about 2, which is not as large as one would expect. What factors in the experiment limit the ratio?

Author: The ratio about 2 must be ascribed to the meta-source used in this work. Detailed discussions are given in Supplementary Information in new section “*Factors affecting the sub-wavelength chiral source*”. In general, we find that the geometric parameters $d, r1, L, l, h$, and load on PZT rings can affect the unidirectional propagation.

4. Fig.2(e). The elliptic polarization of displacement and its direction are well seen for a time interval 2-3 msec. For later time intervals, the polarization seems to be reversed. Please discuss what is being observed at a later time.

Author: Thanks for noticing this question. Before answering it, we have corrected the mistake in the horizontal axis of Figs. 4(c) and (d) (previous Fig. 2(e)) when labelling the time axis in previous figure. However, the phase of u_x and u_y keep the same as before.

The reversed polarization between u_x and u_y comes from the reflected wave **for later time interval after 1.6 msec**. Since we don't use the absorbing boundary condition in experiment, the wave package is reflected by the aluminum plate terminal. The reflected wave propagates in an opposite direction to the original one, so does the polarization direction.

To avoid confusion, we have added the description about this phenomenon in main text as “*In Fig. 4(c), the phase of u_x (blue solid line) at $x = -45$ cm is earlier than the one of u_y (red dash line) before 1.6 msec. Due to wave reflection at the structure terminal, the signal after 1.6 msec gradually contains the information of reflected wave package. The wave vector of reflected wave is opposite with the original one, so as the direction of SAM. Thus, the phase of u_x is gradually later than the one of u_y after 1.6 msec.*”

5. Line 182-185. Please explain why the chiral elastic source on one side of the Al plate generates A0, but not S0 Lamb wave.

Author: In view of this question, we have shown the measured spectrum of u_y in both Figs. 6(a) and (b). The hot spots corresponding to the large value of u_y locate on the A0 branches of Lamb waves, but the S0 components are too slight to be found. As a consequence, A0 mode is dominant in our experimental results. Actually, the dominant S0 wave can hardly be generated when using point-like transducers on one side of aluminum plate. The efficient generation of S0 wave needs elaborately designed transducers, such as specially designed magnetostrictive transducers with high directionality reported in Appl. Phys. Lett. 108, 093501 (2016).

According to this question, we have added in our original text the sentence “*Evidently, the components are very large for A0 wave but very minor for the lowest-order symmetric Lamb (S0) wave, which confirms the efficient generation of A0 wave*”.

6. Please include the derivation of Eq.(2) from the superposition of longitudinal and transverse waves to compose Rayleigh waves.

Author: According to your suggestion, we have added a more detailed derivation of the SAM of Rayleigh wave and the theoretical derivation for Lamb wave in Section Methods.

7. Please include the derivation of Eq.(3) and related references.

Author: According to your suggestion, we have included a more detailed derivation about Eq. 3 (Eq. 2 in revised version).

Reviewer #3:

1、 The difference between intrinsic spin and pseudo-spin should be one of the focuses of main text, but this is only briefly mentioned in abstract and introduction. Not enough. I think it's better to provide comprehensive discussions about the difference between intrinsic spin and pseudo-spin, as well as the advantage of intrinsic elastic spin in application prospect in more details. Most importantly, this is also a good chance (for the convenience of readers) to clarify the importance of the physical elastic spin and how it demarcates from the mathematical pseudo-spin, since the pseudospin and pseudospin-momentum locking have been widely studied in metamaterials, for example the valley pseudo-spin and vortices in Nature Physics 13, 369 (2017), PRB 95, 174106 (2017) and reference therein.

Author: Thanks for your suggestion. As you concerned, the difference between pseudo-spin and spin angular momentum (SAM) is an important aspect. We have discussed the topological nature and the difference between intrinsic spin and pseudo-spin. The intrinsic spin relates to the SAM of elastic wave which have been uncovered from the elastic wave function [PNAS 115, 9951 (2018)]. As for pseudo-spin, it relates more to the effective energy flux in special structures or lattice systems and can be described, mathematically, by the effective Hamiltonian similar to Dirac fermions [Nature Communications 9, 3072 (2018); Nature Physics 13, 369 (2017)]. Meanwhile, the elastic SAM results in nontrivial spin Chern number as $C_{spin} = 4$, provides two pairs of unidirectional edge spin transport with opposite spin, while the topological Z_2 invariant is zero without backscattering immunity [Science 348, 1448 (2015)]. However, the pseudo-spin can provide nontrivial topological Z_2 invariant and result in robust edge modes against backscattering [Science 348, 1448 (2015)]. Moreover, the physical elastic spin provides a new understanding of spin and topological nature of elastic

waves, and a new freedom to control elastic waves, in lacking of delicately designed metamaterials.

According to your suggestion, we have added the explanations in the main text as:

“Different from pseudo-spin, the SAM of elastic wave has been theoretically uncovered from the elastic wave equation in conventional solids and describes the local rotation of displacement polarization [29]. The SAM gives rise to the quantum spin Hall effect (QSHE) for surface wave along the interface of conventional medium with the spin-momentum locking and unidirectional wave transport [1,29]. In contrast, the pseudo-spin relates more to the synthesization of different states like clockwise and anticlockwise energy flux in special structures or lattice systems [27,30]. As such, the edge modes related with pseudo-spin can be topological protected in their own states space. Recently, the spin-based on-chip magnetic devices and sensors [31,32] have been shown to contain spin transfer between homogeneous elastic and magnon systems without meta-structures, indicating that the elastic spin must play a significant role in the magnetic dynamics through elastic-magnetic SAM couplings.”

and

“The existence of spin-dependent edge modes can also be confirmed by the specific spin Chern number of homogeneous solids. Considering a pair of basic elastic wave states, i.e. the normalized longitudinal wave $\mathbf{u}_l = (1,0,0)e^{ik_l x}$ and circular polarized transverse wave $\mathbf{u}_t = (0,i\sigma,1)e^{ik_t x}$ where \mathbf{k}_l and \mathbf{k}_t are wave numbers of longitudinal and transverse waves respectively and $\sigma = \pm 1$. One can obtain the Berry curvatures of these two basic states as

$$F_l = \nabla_{\mathbf{k}} \times [-i(\mathbf{u}_l)^* \cdot (\nabla_{\mathbf{k}})\mathbf{u}_l] = 0 \quad \text{and} \quad F_t^\sigma = \nabla_{\mathbf{k}} \times [-i(\mathbf{u}_t^\sigma)^* \cdot (\nabla_{\mathbf{k}})\mathbf{u}_t^\sigma] = \sigma \frac{k_t}{k^3} \quad \text{respectively.}$$

The topological Chern numbers are $C_l = \frac{1}{2\pi} \oint F_l d^2 \mathbf{k} = 0$ and $C_t^\sigma = \frac{1}{2\pi} \oint F_t^\sigma d^2 \mathbf{k} = 2\sigma$ for the longitudinal and transverse wave, respectively. The total Chern number is $C = C_l + \sum_{\sigma=\pm 1} C_t^\sigma = 0$ due to the time-reversal symmetry, but the spin Chern number of elastic wave is $C_{spin} = C_{spin}^l + C_{spin}^t = 0 + \sum_{\sigma=\pm 1} \sigma C_t^\sigma = 4$, leading to the existence of two pairs of edge modes with opposite spins [29]. Note that, these elastic surface modes are not robust against sharp corners or defects due to their trivial topological Z_2 invariant ($C_{spin}/2 \bmod 2 = 0$). So that, these surface modes are lacking of backscattering immunity. Similar topological discussion is also presented in the quantum spin hall effect of light [1].”

2、 The spin of chiral elastic meta-source is excited by four rods, as shown in Fig.1(b). As this source is not exactly a point source, the properties of source will be associated with the geometry size and wave length, and I assume the source will become invalid when the excitation frequency is high enough. Is that true? If it is, how to determine the upper frequency limit of the source? I believe this is important since, in the application of such a chiral source, the working frequency range must be considered. Moreover, just out of curiosity, is it possible to enlarge the ratio of one-way propagation (possible to be perfect? And how?) shown in Fig.2(c) by optimizing the chiral meta-source, either experimentally or theoretically? Relevant discussions in AIP Advances 6, 085007 (2016) about vortices source may be helpful.

Author: We agree with the reviewer that the source will become invalid when the excitation frequency is high enough. As shown in Supplementary Information “*Factors affecting the sub-wavelength chiral source*”, we find that the geometric parameters d, r_1, L, l, h , and the load on PZT rings can affect the unidirectional propagation.

To determine the upper frequency limit of source, although without an explicit formula due to the too many parameters, we can still determine the upper frequency of source or smallest operation wavelength from several aspects in experiment: first, the most upper limit of valid frequency is limited by the size of inner radius of PZT disk which cannot be as small as possible to generate pulses efficiently. Secondly, the imperfect installations and geometric parameters, in particular the distance between air holes, can weaken a little the meta-source. Thirdly, we operate in the frequency range below the resonant frequency of PZT disk. In all this condition, the distance between air holes is less than one third of the smallest wavelength in experiment in our work. Of course, the unidirectional wave generation can be ameliorated in future by modifying the geometrical parameters of meta-source or the installation process.

We have also added some discussion about the optimize of elastic spin source in Sec. discussion as: “*Note that the present chiral elastic source, by elaborating specific phase on each PZT and rod from delaying signals (Supplementary notes D and E, and [29]), is also generalized to study the acoustic spin [37], selective routing of near-field acoustics [38] and optics [39]. Besides, one may alternatively construct different chiral meta-source, by naturally adjust the underlying meta-structure [40] or multi-beam superposition to cope with different scenarios.*”

3、The content of the current manuscript is very condensed. I'd like to suggest the authors present more details, for example:

a) The experiment setup details of Rayleigh wave are not clearly stated, for instance, the geometry size of the solid plate in Fig.2(a). Especially, the length of the plate along y-axis. According to the demonstration of chiral meta-source in Fig.1 and the eq.2 in methods, I think the theoretical model is based on half-infinite condition, which means that the length of the aluminum plate used in the experiment is required to ensure that the experiment can fit the simulation correctly. Also, the geometry and setup of chiral elastic source used for Rayleigh experiments are not well stated. More experimental details need be included in the revision.

Author: a) According to your suggestion, the size of aluminum plates for both Rayleigh and Lamb wave experiment is given in the Supplementary Information. Meanwhile, we also describe more clearly the geometric parameters of the meta-source in the new supplementary Fig. 1, and state in Sample fabrication section as “*The distance from top (bottom) surface of an aluminum rod to closest surface of aluminum plate is l , the distance from top (bottom) surface of an aluminum rod to closest surface of PZT ring is h , the radii of holes drilled in aluminum block is $r_1 = 2.5$ mm, the radii of aluminum rods is $r = 2.25$ mm, the length of aluminum plate along z axis is $e = 8$ mm, and the distance between the meta-source center and x-z surface at $y = 0$ is $L = 12$ mm.*”

Besides, in main text, we added “*Here, the x-z surface at $y = 0$ is seen as the interface between solid block and air, the length along y-axis is large enough to be seen as a semi-infinite condition.*” In the caption of Fig. 1, and added “*, and the aluminum plate has thickness 120 cm,*

~8 times of the maximum Rayleigh wavelength in this work, along y-axis to ensure the Rayleigh mode propagating on x-z surface at $y = 0$ ” at the 1st sentence of paragraph just below section “*Observation of elastic spin for chiral Rayleigh wave*”.

b) The theoretical results about the elastic spin of Rayleigh and Lamb wave in Method are also very important results, which I strongly suggest move to the main text. It should be helpful to understand the intrinsic elastic spin and how it is tightly locked to the directionality. Moreover, the numerical simulation of Rayleigh wave spin and its spin-momentum locking should be helpful for the readers to understand the experimental measurements. Therefore, I suggest the authors move this part from the supplementary material to the main text.

Author: we agree with reviewer that the theory and numerical simulation are important for readers to understand the experimental measurement, and to capture the intrinsic elastic spin locking of Rayleigh and Lamb waves. We have made the following changes. First, the new Eq. 2 is moved to the main text to highlight the intrinsic spin locking of Rayleigh wave. Secondly, the numerical simulation is added to the revised main text together with the theoretical part in Fig.2(a). As such, readers can capture the principal part of theory, and repeat the numerical results even with a computer not so expensive.

Minors:

1、 When mention the surface in main text, please mark the surface with axis (surface x-z, e.g.) to avoid misunderstanding.

Author: We have marked the related surface with axis as “x-z surface”.

2、 It’s better to provide the vibrating phase of each rod in the left panel of Fig.1(b).

Author: We have added the vibrating phase of each rod as $(0, \pi/2, \pi, \pi/2)$ for rods 1-4.

3、 I think “measured spectrum” is a more proper title of Fig.2(d), 3(a) and 3(h).

Author: We have revised the title of these figures according to your suggestion.

4、 I recommend that the style of Fig. 3(b) and (g) remain consistent with the style of Fig. 2(e).

Author: We have revised style of Figs. 6(c) and (d) (previous Fig. 3(b) and (g)) remaining consistent with the style of Figs. 4(c) and (d) (previous Fig. 2(e)).

REVIEWERS' COMMENTS

Reviewer #1 (Remarks to the Author):

The authors have address all of the concerns and questions raised by the Referees. I now recommend the paper for publication.

Reviewer #2 (Remarks to the Author):

The author of the manuscript replied to my questions and suggestions raised in my first report by including additional figures, explanations in main script and method. I believe that made the manuscript much improved and readable by the general audience. I do not have further comments and I suggest the manuscript to be published in Nature Communications.

Reviewer #3 (Remarks to the Author):

I am satisfied with the revisons made by the authors. It can be published as it is.